# A Precision Therapy Approach for Retinitis Pigmentosa 11 Using Splice-Switching Antisense Oligonucleotides to Restore the Open Reading Frame of PRPF31

**DOI:** 10.3390/ijms25063391

**Published:** 2024-03-16

**Authors:** Janya Grainok, Ianthe L. Pitout, Fred K. Chen, Samuel McLenachan, Rachael C. Heath Jeffery, Chalermchai Mitrpant, Sue Fletcher

**Affiliations:** 1Department of Biochemistry, Faculty of Medicine Siriraj Hospital, Mahidol University, Bangkok 10700, Thailand; jgrainok@gmail.com (J.G.); chalermchai.mit@mahidol.ac.th (C.M.); 2Health Futures Institute, Murdoch University, Murdoch, WA 6150, Australia; i.pitout@murdoch.edu.au; 3Centre for Ophthalmology and Visual Science, University of Western Australia, Nedlands, WA 6009, Australia; fred.chen@lei.org.au (F.K.C.); samuel.mclenachan@lei.org.au (S.M.); rachael.heathjeffery@lei.org.au (R.C.H.J.); 4Lions Eye Institute, Nedlands, WA 6009, Australia; 5Department of Ophthalmology, Royal Perth Hospital, Perth, WA 6000, Australia; 6Department of Surgery, University of Melbourne, East Melbourne, VIC 3002, Australia

**Keywords:** retinitis pigmentosa, splicing factor, antisense oligonucleotides, exon skipping, precision medicine

## Abstract

Retinitis pigmentosa 11 is an untreatable, dominantly inherited retinal disease caused by heterozygous mutations in pre-mRNA processing factor 31 *PRPF31*. The expression level of *PRPF31* is linked to incomplete penetrance in affected families; mutation carriers with higher PRPF31 expression can remain asymptomatic. The current study explores an antisense oligonucleotide exon skipping strategy to treat RP11 caused by truncating mutations within *PRPF31* exon 12 since it does not appear to encode any domains essential for PRPF31 protein function. Cells derived from a patient carrying a *PRPF31* 1205C>A nonsense mutation were investigated; *PRPF31* transcripts encoded by the 1205C>A allele were undetectable due to nonsense-mediated mRNA decay, resulting in a 46% reduction in *PRPF31* mRNA, relative to healthy donor cells. Antisense oligonucleotide-induced skipping of exon 12 rescued the open reading frame with consequent 1.7-fold *PRPF31* mRNA upregulation in the RP11 patient fibroblasts. The level of *PRPF31* upregulation met the predicted therapeutic threshold of expression inferred in a non-penetrant carrier family member harbouring the same mutation. This study demonstrated increased *PRPF31* expression and retention of the nuclear translocation capability for the induced PRPF31 isoform. Future studies should evaluate the function of the induced PRPF31 protein on pre-mRNA splicing in retinal cells to validate the therapeutic approach for amenable RP11-causing mutations.

## 1. Introduction

Retinitis pigmentosa 11 (RP11, OMIM 600138) is an autosomal dominant inherited retinal disease accounting for up to 10% of all autosomal dominant RP [1,2]. The age of onset of the first symptom presentation varies between 6 and 71 years [3,4], with the first symptom of night blindness commonly occurring within the first decade of life [5]. RP11 is caused by heterozygous mutations in the pre-mRNA processing factor 31 (*PRPF31*) gene. To date, approximately 613 unique variants distributed throughout the *PRPF31* have been reported (ClinVar), with no effective therapies available.

The *PRPF31* is located on chromosome 19q13.4, whose canonical isoform comprises 14 exons that encode 499 amino acids. PRPF31 is a ubiquitously expressed splicing factor that functions in spliceosome assembly. Although splicing factors are essential in most cells, mutations in *PRPF31* and other splicing factor genes do not impair splicing in all tissues and are currently known to only affect the retina. Restriction of symptoms to the retina may reflect the exceptionally high demand for mRNA processing, particularly in photoreceptors and retinal pigment epithelium (RPE), and the reliance on alternative splicing in these cell types [6,7] suggesting a critical role of pre-mRNA splicing in retinal homeostasis and pathogenesis in retinal dystrophy. The mechanism leading to the disease phenotypes for *PRPF31*-associated RP appears to be predominantly haploinsufficiency [8,9]. However, the precise mechanisms leading to RP11 phenotypes remain largely unknown.

Incomplete disease penetrance has been observed within RP11-affected families, which has been correlated to the expression level of *PRPF31* [4]. Family members with higher PRPF31 expression were reported to be asymptomatic, while those members expressing lower levels of PRPF31 presented with an RP11 phenotype. Several genetic modifiers were reported to influence the regulation of *PRPF31* expression at the transcription level, including CCR4-NOT transcription complex subunit 3 (CNOT3), *MSR1 repeat number,* and expression quantitative trait loci (eQTLs) on Chr.14q21-23 [10,11,12,13]. Although the precise mechanism of disease penetrance is largely unknown, the expression level of PRPF31 in non-penetrant cases may guide the disease correction threshold required for therapeutic interventions designed to increase the expression of PRPF31. A detailed investigation of PRPF31 expression in RP11-affected families is therefore a worthwhile approach to determining intervention strategies for the currently untreatable RP11 disease.

Since the pathogenesis of RP11 is proposed to be *PRPF31* haploinsufficiency, gene replacement, gene correction (gene editing), and gene augmentation through increased *PRPF31* expression from the unaffected allele could offer a potential treatment for RP11. Strategies aimed at enhancing PRPF31 expression involve targeting the disease modifier gene, *CNOT3*, to indirectly increase *PRPF31* expression from the healthy allele [14,15], or gene augmentation through adeno-associated virus (AAV) delivery of functional *PRPF31* [16]. In contrast, our approach offers an alternative by directly targeting *PRPF31* with splice-switching antisense oligonucleotides (AOs), allowing *PRPF31* expression to remain under endogenous regulation.

Splice-switching AOs are short synthetic, chemically modified nucleic acid molecules, typically 15–30 residues in length, that are capable of complementary binding to mRNA thereby altering splice site selection [17,18]. Several splice-switching AOs have been successfully developed for the treatment of fatal muscle-wasting disorders, Duchenne muscular dystrophy (DMD) and spinal muscular atrophy [19,20]. Three drugs were granted accelerated approval by the Food and Drug Administration in 2016 and later in 2019 and 2021 for skipping DMD exon 51 (*Exondys51* or *Eteplirsen*) [21], exon 53 (*Vyondys53* or *Golodirsen*) [22], and exon 45 (*Amondys45* or *Casimersen*) [23,24], respectively. All three drugs exploit the same concept of restoring the open reading frame around a frameshifting deletion of one or more exons to produce an internally truncated protein that retains partial function. Currently, investigations of splice-switching AOs as potential therapeutics for several genetic diseases are underway [25,26,27].

The present study aims to extend the application of splice-switching AOs to treat a currently untreatable haploinsufficiency disease with moderate prevalence and characterized by incomplete penetrance. We report a potential therapeutic intervention for selected mutations causing the inherited retinal disease, RP11, by incorporating insights into the relative expression levels of *PRPF31* and functional domains of the PRPF31 protein. The expression of *PRPF31* in RP11 patients and a non-penetrant case was explored to inform the level of upregulation likely to be required for meaningful benefits. A targeted therapeutic strategy for RP11, caused by a *PRPF31* nonsense mutation in exon 12, was developed using splice-switching AOs to mediate the skipping of the exon harboring the premature termination codon (PTC). The exclusion of *PRPF31* exon 12 can remove the PTC and maintain the open reading frame of the transcript. The truncated PRPF31 protein could retain partial function since exon 12 does not encode any known functional domain. The results from this study provide a proof-of-concept for effective AO-induced isoform switching to rescue the expression of the *PRPF31* disease-associated variant transcript in RP11 cells from being degraded through nonsense-mediated mRNA decay.

## 2. Results

### 2.1. PRPF31 Level Is Linked to Incomplete Disease Penetrance in an RP11 Family with PRPF31 c.1205 C>A Mutation

This study characterized the expression level of *PRPF31* in cells derived from a patient with RP11 as a consequence of *PRPF31* c.1205 C>A (Ser402*). Figure 1a shows the pedigree of the RP11 family with incomplete disease penetrance arising from a nonsense mutation that introduces a PTC within *PRPF31* exon 12. Samples from six symptomatic RP11 subjects, III2, III5, III12, III6, IV8, and IV9 (aged between 19 and 41 years), and a non-penetrant family member, II4 (57-year-old), from the single pedigree were included in this study. Non-familial fibroblast lines served as healthy controls.

Polymerase chain reaction (PCR) primers were designed to amplify genomic DNA across *PRPF31* exon 12, and the amplicons underwent Sanger sequencing to confirm the heterozygous mutation c.1205 C>A in a representative RP11 case III6 and in the non-penetrant family member II4, whereas the sequences of the healthy control samples were consistent with the reference sequence (Figure 1b). Quantitative reverse transcription polymerase chain reaction (RT-qPCR) was used to quantify the expression of *PRPF31* transcripts in extracts from RP11 patients and the non-penetrant individual’s cells and compare them to those of healthy controls. The results showed a 46% reduction in *PRPF31* expression in RP11 patients’ cells and a 34% reduction in the non-penetrant family member, relative to healthy control fibroblasts (Figure 1c).

Since RP11 disease phenotypes are restricted to the retina, the level of *PRPF31* expression in a retinal-specific cell type is here compared to skin fibroblasts (‘proxy’ cells) to establish a proof-of-concept AO screening study. Retinal pigment epithelium (RPE) differentiated from RP11 patient-derived induced pluripotent stem cells (iPSC-derived RPE) is reported to have pre-mRNA splicing function deficits due to PRPF31 insufficiency [28]. Therefore, the correlation of *PRPF31* levels between the non-penetrant mutation carrier and a healthy control family member was further validated in iPSC-derived RPE to assess *PRPF31* expression levels in these proxy cells. The relative expression of *PRPF31* in RP11, non-penetrant, and healthy control iPSC-derived RPE remained consistent between iPSC-derived RPE and primary fibroblasts (Figure 1c,d), indicating the potential use of fibroblasts derived from RP11 patients as proxy cells for initial drug screening in vitro. These findings further support *PRPF31* haploinsufficiency as the primary mechanism underlying disease phenotypes, with incomplete penetrance in the RP11 family studied here, suggesting that even a subtle increase in *PRPF31* levels in the retina of RP11 subjects may suffice to provide functional benefits in mRNA splicing.

### 2.2. PRPF31 Transcript Harboring c.1205 C>A Mutation Is Highly Susceptible to mRNA Degradation via Nonsense-Mediated mRNA Decay

Transcripts bearing a PTC are generally degraded by the nonsense-mediated mRNA decay (NMD) pathway. Nonsense-mediated decay of a mammalian transcript may be predicted by applying the 50–55-nucleotide rule. Generally, if the premature stop codon lies more than 50 bases upstream of the nearest exon junction complex (EJC), then the transcript is sensitive to NMD [29].

The *PRPF31* c.1205 C>A mutation is annotated to create a PTC in exon 12. This PTC is 71 bases upstream of the EJC and should render the affected transcript susceptible to degradation through NMD. No transcript carrying the *PRPF31* c.1205 C>A variant was amplified from the RP11 patient-derived RNA. We further assessed the expression and the susceptibility of this transcript to nonsense-mediated mRNA decay using an NMD inhibitor (NMDI14). After patient fibroblasts were treated with the NMD inhibitor, the *PRPF31* transcripts increased in a dose-dependent manner, as determined using semi-quantitative RT-PCR (Figure 2a,b), suggesting substantial degradation of the transcripts encoded by the disease-associated allele in patient fibroblasts, due to NMD. Sanger sequencing of the amplicons confirmed the presence of the *PRPF31* c.1205 C>A transcript in the absence of NMD.

### 2.3. Strategic Design of Antisense Oligonucleotide to Bypass a Premature Termination Codon in PRPF31 Exon 12

AO-mediated splice site selection is a promising approach to alter transcript abundance or structure to overcome human disease. Exon skipping can restore the open reading frame to treat multiple genetic diseases, including Usher syndrome (*USH2A*) [30], and Duchenne muscular dystrophy (*DMD*) [31]. The latter has demonstrated clinical benefits after PMO treatment [32]. In the present study, the application of AO-induced exon skipping is explored in RP11 patient cells carrying a heterozygous *PRPF31* c.1205 C>A variant. This mutation results in a premature termination codon (PTC) in exon 12 that truncates protein synthesis. The exclusion of *PRPF31* exon 12 from the mature transcript is hypothesized to bypass the PTC, maintain the open reading frame of the transcript, and consequently produce a modified PRPF31 protein.

The *PRPF31* transcript comprises 14 exons that, when processed, encode 499 amino acids essential for the stabilization of the U4/U6-U5 tri-snRNP complex during pre-mRNA splicing [33]. Functional domains of RP11 include the NOSIC domain (named after the central domain of ‘Nop56/SIK1-like protein’), nucleolar protein (Nop) domain, and nuclear localization signal (NLS) encoded by exons 4–6, 7–10, and 10–11, respectively (Figure 2a). A key consideration when using an AO-induced exon skipping approach in the current study is to minimize interruption of the essential functional domains of the target protein.

Another consideration for the exon skipping approach is to maintain the open reading frame of the target transcript in order to prevent an out-of-frame transcript and consequent mRNA degradation. Figure 3a indicates the reading frame of the canonical *PRPF31* transcript represented as follows: in-frame exons are depicted by rectangles, whereas exons with junctions that interrupt codons are shown with chevron sides. Figure 3b shows the AO design strategy to induce the exclusion of exon 12 harboring the PTC from the *PRPF31* transcript to generate an internally truncated *PRPF31* isoform missing 129 nucleotides. Excision of exon 12 from the *PRFP31* transcript does not disrupt the reading frame and therefore results in the production of an internally truncated PRPF31 isoform, missing 43 amino acids that are not known to comprise any functional domain.

### 2.4. Screening of Splice-Switching AO to Mediate PRPF31 Exon 12 Exclusion and Restore the Open Reading Frame in Fibroblasts Derived from an RP11 Patient

Five splice-switching AOs were designed as 25-mers to target exonic splice enhancer (ESE) motifs in *PRPF31* exon 12 or donor/acceptor splice sites to inhibit exon recognition during pre-mRNA splicing (sequences listed in Table 1). The ESE motifs were predicted using the following online tools: Human Splice Finder version 3.1: http://www.umd.be/HSF/ (accessed on 20 June 2019) and SpliceAid: http://www.introni.it/splicing.html (accessed on 20 June 2019). Predicted ESE motifs and target regions of each AO along the exon are presented in Figure 4a. The splice-switching AOs were synthesized as chemically modified 2′*O*-methyl bases on a phosphorothioate backbone.

Splice-switching AOs were transfected into primary fibroblasts derived from an RP11 patient carrying a *PRPF31* c.1205 C>A mutation at concentrations of 25 nM and 50 nM to screen for a sequence that efficiently induces specific skipping of *PRPF31* exon 12. RT-PCR was used to assess AO-mediated exon 12 exclusion by amplifying across exons 10–13, generating an amplicon 129 bp smaller than the full-length transcript product (384 bp). The results from this study showed that the effectiveness of splice-switching AO-induced exon skipping was not strongly influenced by the presence of ESE motifs within the AO target region. Among all splice-switching AO sequences tested, AO4 was found to be the most efficient at mediating the exclusion of *PRPF31* exon 12 from the mature transcript with up to 89% exon 12 skipping achieved, despite low predicted splice enhancer scores (Figure 4b,c). AO1 and AO2 that target regions with high splice enhancer scores induced low levels of exon 12 skipping. The sequences AO3 and AO5 target the central region of the exon and are predicted to have no potential for splice enhancers and the donor splice site with a moderate splice enhancer score, respectively, and they did not induce exon 12 skipping. Sanger sequencing of the induced shorter RT-PCR products confirmed precise exon 12 skipping and the exon 11/13 junction (Figure 4d). Therefore, due to efficient exon 12 skipping without interference from neighboring exons, AO4 was selected as a lead AO sequence for further evaluation.

### 2.5. PMO-Induced Exclusion of Exon 12 Bypasses the Disease-Causing PTC and Rescues PRPF31 Expression

The splice-switching AO4 sequence was synthesized as a PMO (so-called PMO4); a chemistry that has shown excellent clinical tolerability and is used in approved therapeutics [32,34,35]. The PMO was transfected into patient fibroblasts using a NEON transfection system and incubated for 7 days. A PMO control sequence (Gene Tools control; GTC) that does not alter any transcripts was used as a sham control treatment. RT-PCR analysis showed that the PMO effectively induced the excision of *PRPF31* exon 12 in a dose-dependent manner in RP11 fibroblasts (Figure 5a). Furthermore, the removal of exon 12, harboring the c.1205 C>A PTC from the transcript, is anticipated to maintain the open reading frame and transcript integrity. Quantitative RT-PCR was used to assess the expression of total *PRPF31* transcript in PMO-treated patient fibroblasts. The results revealed that total *PRPF31* mRNA was upregulated in a dose-dependent manner with up to a 1.70-fold increase achieved, compared to the control PMO-treated and untreated control cells (Figure 5b).

Further investigation considered whether an increase in the *PRPF31* mRNA level observed in Figure 5b originated equally from both full-length and truncated transcripts or an excess of either transcript. Two different qPCR primers were designed to distinguish between full-length and internally truncated transcripts. qPCR primers explicitly designed to anneal to exons 3/4 were expected to amplify total *PRPF31* transcripts arising from both full-length and ∆exon 12 *PRPF31.* By contrast, the primers designed specifically to anneal to the exon 12/13 junction were expected to amplify only the full-length *PRPF31* transcript. The expression of ∆exon12 *PRPF31* was calculated from the differences between the level of total *PRPF31* and full-length *PRPF31* expression. The results demonstrated the increase in *PRPF31* transcripts lacking exon 12, while full-length transcripts remained nearly unchanged when compared to controls (Figure 5c). The results from this study suggest that the PMO effectively mediates exon 12 skipping to bypass the PTC and rescue the mutant allele, which otherwise would be degraded via NMD.

To explore why PMO4 exhibited a stronger tendency to skip exon 12 in the mutant transcript compared to the healthy allele, an analysis was conducted on alterations in splicing motifs and the secondary structure of PRPF31 mRNA. The prediction of splicing motifs within PRPF31 exon 12 indicated that the mutation at c.1205 did not affect splicing strength (refer to Figure 4a). Using an online tool (RNAfold), predictions were generated for the secondary structure of exon 12 and its flanking intron regions (50 nucleotides on each end). As demonstrated in Figure 5d,e, a slight elevation in the minimum free energy was determined in the mutant allele compared to the healthy allele, with a lower minimum free energy state suggesting greater stability of the latter structure.

While the general secondary structure of the region stayed mostly consistent, the simulations indicated a significant alteration in the secondary mRNA structure at the PMO4 binding site within the mutant allele, resulting in the formation of a stem-loop. This mutation-induced alteration potentially influences the secondary RNA structure, indicating the capability of a single base change to cause such modifications. The predicted altered structure might facilitate increased accessibility for PMO binding, thereby raising the probability of PMO4-mutant RNA binding relative to PMO4-healthy RNA binding and subsequently impacting the efficiency of exon skipping.

### 2.6. Truncated PRPF31 Protein Retains the Ability to Translocate into the Nucleus

The exclusion of exon 12 from the transcript is expected to direct the synthesis of an internally truncated PRPF31 protein, missing 43 amino acids, that may retain function, since exon 12 does not encode any known functional domain(s), in effect generating a hypomorphic protein product. The majority of previous studies on exon skipping were applied to large genes, with the removal of less than 10% of the amino acids without detrimental effects on protein conformation [33,34]. However, the effect of excluding 43 amino acids out of the 499 amino acids on PRPF31 protein conformation is unknown, considering that the neighboring exon 11 encodes the critically important nuclear localization signal for translocation of the PRPF31 protein into the nucleus. Wheway et al. [36] reported that some missense mutations in PRPF31 that alter protein conformation led to protein aggregation or interruption of the nuclear translocation capability of the altered protein. In the current study, the alternative PRPF31 protein isoform induced by skipping of exon 12 was examined to ascertain if the loss of these 43 amino acids altered the solubility or localization of the protein to the nucleus.

The expression and localization of PRPF31 in healthy fibroblasts and patient fibroblasts harboring the nonsense mutation were assessed by immunofluorescent staining. The PRPF31 protein was localized in the nucleus of the RP11 patient fibroblasts, while the protein expression level was reduced relative to the healthy control, as shown in Figure 6. When the RP11 cells were treated with PMO4, the truncated protein remained localized in the nucleus of RP11 cells, with a moderate enhancement of PRPF31 expression. Importantly, aberrant PRPF31 protein accumulation or cytoplasmic localization due to the production of the alternative PRPF31 isoform was not observed in this study. The results suggest that the domain encoded by *PRPF31* exon 12 may be partially dispensable and that the exon could be removed from the transcript while allowing the induced PRPF31 isoform to retain some function.

## 3. Discussion

Loss-of-function mutations in *PRPF31* are a significant cause of autosomal dominant RP, leading to vision impairment, with no currently available therapy. Retinal function defects caused by PRPF31 mutations include fewer and shorter cilia, reduced phagocytosis, impaired RPE tight junctions, abnormal cytokine secretion, and incomplete splicing of numerous retinal transcripts [37]. Recently, gene editing using CRISPR-Cas9 showed rescue of PRPF31 functions in patient-derived retinal cells and organoids [37]. Splice-switching AOs are capable of modifying pre-mRNA splicing to alter transcript isoforms for translation into putatively functional proteins and subsequent reduction of disease progression and severity (for review see [38,39]). Clinical benefits have been seen in DMD after PMO treatment [32]. The applications of splice-switching AOs are here extended to an inherited retinal degenerative disease, RP11, caused by a *PRPF31* nonsense mutation.

The age of onset of RP11 disease varies between patients, even within the same families, with a median age within the 3rd and 4th decades of life [4]. Nonsense and frameshift mutations are more commonly detected in younger patients, with the first symptoms of night blindness occurring between 8 and 12 years of age [36]. In contrast, older patients typically present with in-frame and missense mutations and show first symptoms of night blindness at about 27 years [36]. An in-frame deletion within exon 6 was, however, reported to result in onset at an age of over 70 years, clearly indicating that some exons are not essential for near-normal function [3]. The accumulated data suggest a correlation between a severe reduction in *PRPF31* levels as the result of nonsense mutations, with early symptom development. This study explores the conversion of a PTC allele-mediated loss-of-function to a less severe internally truncated, in-frame *PRPF31* transcript with the potential to reduce disease progression in RP11.

The *PRPF31* gene includes 14 exons that encode a protein of 499 amino acids that function in pre-mRNA splicing in all cell types. The functional domains of PRPF31 include the Nop domain and nuclear localization sequence that are encoded by exons 7–10 and exons 10–11, respectively. To the best of our knowledge, no pathogenic deletions of, or mutations causing, *PRPF31* exon 12 skipping have been reported to date.

In the current study, splice-switching AOs capable of facilitating *PRPF31* exon 12 skipping from the mRNA were designed as a potential therapeutic intervention to bypass the PTC caused by a heterozygous PRPF31 c.1205 C>A mutation. Five splice-switching AOs were designed to prevent exon 12 recognition, and the results from this study indicate no obvious correlation between the efficiency of the splice-switching AOs and the predicted ESE strength scores. The splice-switching AOs targeting canonical splice sites were not effective, with an exon skipping efficiency of less than 10%. A similar outcome was reported during the development of DMD exon 51 skipping AOs by V. Arechavala-Gomeza et al. [31].

The lead candidate AO4 efficiently induced exon 12 exclusion from the transcript after transfection with sequences as either the 2′*O*-methyl PS or PMO chemistries. In addition, the lead AO identified in this study may be applicable to addressing other mutations in exon 12 reported elsewhere [40,41,42,43,44]. Patient fibroblasts treated with the optimized splice-switching AO result in increased *PRPF31* transcript levels and the production of a truncated protein that localizes to the nucleus. Neither protein aggregation, mislocalization, or cell death were evident after exon 12 skipping was induced at nearly 100%. As demonstrated in this study, the *PRPF31* c.1205 C>A change rendered the transcript highly susceptible to NMD. It was further determined that the patient fibroblasts carrying this mutation expressed *PRPF31* mRNA at about half of the level of that found in the healthy control cells. This reduction in *PRPF31* observed in RP11 patients was slightly greater than that observed in a non-penetrant family member harboring the same mutation, both in fibroblasts and iPSC-derived RPE. A small increase in *PRPF31* is therefore expected to reach the therapeutic threshold predicted from the evaluation of PRPF31 expression in non-penetrant mutation carriers.

In this study, a 1.70-fold increase in *PRPF31* expression was observed after skipping exon 12 was induced by splice-switching AOs in patient fibroblasts. Splice-switching AOs may be expected to affect both the disease-associated variant and normal transcripts if the splice-switching AO anneals to both alleles with equal efficiency. This study showed that AO4 preferentially induced exon 12 exclusion from the mutated allele transcript carrying the PTC, at lower transfection concentrations. However, when inducing a high level of skipping (>80%), exon 12 was also excluded from the normal healthy allele transcript and expression of the full-length transcript was reduced (data not shown). Differential skipping of the target exon from the transcripts encoded by the two alleles may be due to alteration of the ESE or ESS motifs in the mutation-bearing allele by the single base substitution and could therefore influence exon recognition. Subtle changes in RNA conformation of the mutated allele close to the AO binding site were predicted using free energy minimization online software that suggests greater AO accessibility and therefore exon skipping. Indeed, the lead candidate AO4 annealed to the mRNA targets 11 bases downstream of the mutation site. This phenomenon could be beneficial to strategies designed to rescue other disease-causing alleles and produce a truncated protein with partial function while retaining the capacity of the remaining normal healthy allele to yield a full-length protein product. Alternatively, this strategy may be applied to allele-specific splice-switching for dominant-negative diseases, with minimal disruption of the normal allele. Currently, allele-specific AO therapeutics primarily rely upon RNase H activation and siRNA, leading to transcript knockdown.

Haploinsufficiency diseases, while dominantly inherited, are potentially treatable by various gene augmentation strategies. Approaches to the treatment of RP11 include boosting *PRPF31* expression from the healthy allele, gene editing, and gene replacement, (for review see Aweidah 2023 [45]). Significant improvements in gene expression and enhanced cellular characteristics in patient-derived models are reported [37]. Due to the *PRPF31* mutation heterogeneity, mutation-agnostic approaches such as AAV gene therapy have been explored for direct *PRPF31* augmentation, showing some phenotype restoration in mouse models [16]. However, AAV injection (usually subretinal) presents challenges with limited biodistribution to photoreceptor cells and RPE, the crucial target cells in the treatment of RP11. In addition, poor translatability of preclinical and clinical trials of AAV gene therapy in inherited retinal diseases has limited clinical implementation of these therapies [46,47]. Alternatively, a peptide-conjugated PMO to target *CNOT3*, a disease modifier of RP11, for the upregulation of PRPF31 demonstrated promising results in patient-derived retinal cells and is currently in a Phase I clinical trial [14,15].

Considering the slow progression of RP11, it is essential to develop potential treatments that have prolonged effects to continuously prevent vision loss. AOs present a promising drug class for managing RP11 due to their relatively long half-life [48]. We present a precision approach using AOs to address *PRPF31* mutation within exon 12 as a proof-of-concept for potential use in a subset of RP11 cases. This approach holds promise for other cases in which selected mutation-bearing exons, not encoding critical functional domains, could be targeted [49,50,51]. Currently, several FDA-approved drugs exploit this strategy for the treatment of Duchenne muscular dystrophy (DMD), including *Eteplirsen, Golodirsen, Casimersen,* and *Viltolarsen* [21,22,23,24,52,53], designed to skip selected exons from the *DMD* transcript. These exon skipping strategies share a common concept: restoring the open reading frame, thereby enabling the production of a truncated dystrophin protein with retained functionality, resulting in observed clinical benefits. While the focus in the present study was on investigating AO-induced skipping of *PRPF31* exon 12 to overcome the PTC and restore the open reading frame, this approach may also apply to other in-frame exons within the gene, including exon 4. Although *PRPF31* exon 13 is ‘in-frame’ and the encoded protein region does not have a predicted function, it does contain multiple phosphorylation sites essential for PRPF31 function and is therefore not suitable for an exon skipping strategy [54].

Further investigations into the activity of the induced PRPF31 isoform are required to determine the feasibility of our exon skipping approach, which is a limitation of the present study. Since patients with *PRPF31* mutations manifest the disease phenotypes only in the retina, splice-switching AO-induced exon 12 exclusion should be further evaluated for functional improvement in *PRPF31*-associated RP and photoreceptor cell models, such as iPSC-derived neuroretinal organoids and retinal pigment epithelium. Due to limitations in the transfection of oligonucleotides to retinal tissues, a highly efficient transfection method, e.g., peptide-conjugated PMO is required to effectively mediate *PRPF31* exon 12 skipping to confirm the function of the truncated PRPF31 protein isoform in the retina. In addition to evaluating efficacy, it is important to consider the potential off-target effects of antisense oligonucleotides. These effects are influenced by the chemical modifications to the nucleobases and backbone; however, careful design and the highly specific base pairing with target mRNA can limit off-target hybridization. A comprehensive review of pre-clinical assessments of AOs has been thoroughly conducted by Goyenvalle et al. [55].

In conclusion, the application of splice-switching AOs was extended to inherited retinal disease caused by a specific *PRPF31* mutation by targeting the removal of *PRPF31* exon 12 that harbors a PTC and restoring the open reading frame during pre-mRNA processing. The splice-switching AO effectively mediated the skipping of exon 12 to bypass the PTC and rescue the transcript, which is otherwise degraded via the NMD pathway. A dose-dependent upregulation of endogenous *PRPF31* was observed at a sufficient and controllable level to reach a likely therapeutic threshold of expression. The resulting truncated PRPF31 protein appears to retain the ability to translocate to the nucleus similarly to PRPF31 healthy cells, with no evidence of protein aggregation.

## 4. Materials and Methods

### 4.1. Dermal Fibroblast Culture

A group of RP11 patients and a non-penetrant family member carry a heterozygous *PRPF31* c.1205 C>A nonsense mutation as previously described [56]. Fibroblasts were maintained in Dulbecco’s Modified Eagle Medium (DMEM) supplemented with 10% FCS, at 37 °C, and 5% CO_2_ in a humidified incubator. Cells were passaged approximately 2–3 times per week according to a standard trypsinization protocol.

### 4.2. iPSC-RPE Generation

Induced pluripotent stem cell (iPSC) lines from a patient with dominant *PRPF31* mutation and a related non-penetrant carrier were generated as described in McLenachan et al. [57]. Human iPSC colonies were dissociated into small pieces by mechanical dissociation after 2–4 min of incubation with EDTA buffer (0.5 mM EDTA and 30 mM NaCl in Dulbecco’s phosphate-buffered saline without calcium or magnesium). Cell clusters were seeded into suspension culture plates and cultured in DMEM/F-12 medium (ThermoFisher, Scientific, Scoresby, VIC, Australia) containing MEM Non-Essential Amino Acids Solution (NEAA, ThermoFisher Scientific, Scoresby, VIC, Australia), B27 (ThermoFisher, Waltham, MA, USA), human recombinant IGF-1 (Stem Cell Technologies, Melbourne, VIC, Australia), and KnockOut Serum Replacement (KOSR, ThermoFisher Scientific, Scoresby, VIC, Australia). The KOSR concentration was gradually decreased from 20% for the first 5 days to 15% until day 12 and to 10% until day 35. From day 35, KOSR was removed from the media. Half media changes were performed every day, with full media changes every second–third day, up to 200 days of culture. RPE organoids were removed from RO cultures after 6–8 weeks of differentiation and transferred into geltrex-coated 24-well plates with RPE media containing DMEM/F-12 medium with 1× NEAA, 1× B27, human recombinant 10 ng/mL IGF-1 (Stem Cell Technologies), and 4% KOSR (ThermoFisher Scientific, Scoresby, VIC, Australia). Media were changed every third day. The RPE organoids formed pigmented cell clusters within 4–6 weeks (RPEp0). RPE clusters were dissociated by incubation with TrypLE Express Enzyme (ThermoFisher Scientific, Scoresby, VIC, Australia) for 20 min at 37 °C and passaged into new Geltrex^TM^ (ThermoFisher Scientific, Scoresby, VIC, Australia) -coated wells using a split ratio of 1:5. Pure RPE monolayers were usually established after two passages. RPE monolayers were used for experiments at passage 3–4, 4 weeks after plating.

### 4.3. Nonsense-Mediated Decay Inhibitor Treatment

NMDI14, a nonsense-mediated decay (NMD) inhibitor substance (Sigma-Aldrich, Bayswater, VIC, Australia), was dissolved in an EtOH/DMSO solution (1:1 ratio) at a concentration of 50 mM. Patient fibroblasts were treated with 20 μM NMD14 and incubated for 16 h in an OptiMEM culture medium. Cells were harvested for RNA extraction.

### 4.4. Antisense Oligonucleotide Design and Synthesis

Antisense oligonucleotides were designed to anneal to splice enhancer motifs, as predicted by the online tools: SpliceAid and Human Splice Finder version 3.1. The *PRPF31* exon nomenclature was determined with respect to the NCBI reference sequence NM_015629.4. Initial screening of AO sequences was performed using 2′-*O*-methyl modified bases on a phosphorothioate backbone (PS). The 2′*O*-methyl PS AOs were purchased from TriLink Biotechnologies (San Diego, CA, USA). Following the identification of an optimal 2′*O*-methyl PS sequence, it was synthesized as a phosphorodiamidate morpholino oligomer (PMO). The PMOs were purchased from GeneTools LLC (Philomath, OR, USA).

### 4.5. AO Transfection

For initial AO sequence screening, fibroblasts were seeded at the density of 1.5 × 10^4^/well in a 24-well plate and incubated overnight prior to approximately 70–90% cell confluence, prior to 2′*O*-methyl PS AO transfection. The 2′-*O*-methyl PS AOs, which were diluted in OptiMEM (Cat#31985062, Life Technologies, ThermoFisher Scientific, Scoresby, VIC, Australia), were transfected into fibroblasts using Lipofectamine 3000 Transfection Reagent (Cat#L3000015, ThermoFisher Scientific, Scoresby, VIC, Australia), according to the manufacturer’s protocol, at a molar ratio of transfection reagent to splice-switching AO of 2:1 and incubated for 24–48 h in OptiMEM (ThermoFisher Scientific, Scoresby, VIC, Australia). A splice-switching AO that targets *SMN1* and induces exon 7 skipping was used as a positive transfection control [58]. The splice-switching AO sequence that induced the most efficient skipping of each targeted exon was then synthesized as a PMO.

Transfection of PMOs into patient fibroblasts was achieved using the Neon^TM^ Transfection System (Cat#MPK5000, ThermoFisher Scientific, Scoresby, VIC, Australia), according to the manufacturer’s protocol, at concentrations of 50 µM, 20 µM, and 5 µM/5 × 10^5^ cells in a 6-well plate, with a transfection setting of 1650 volts and 10 ms for 3 pulses. A PMO sequence that does not anneal to any known transcript (‘GeneTools Control’) was used as a sham treatment and is commercially available as a PMO from Gene Tools LLC (Philomath, OR, USA). PMO-transfected cells were incubated for 7 days prior to RNA extraction.

### 4.6. RNA Purification and RT-PCR

Total RNA was isolated from the cells using the MagMAX™-96 Total RNA Isolation Kit (Cat#AM1830, ThermoFisher Scientific, Scoresby, VIC, Australia), according to the manufacturer’s protocol. The concentration and purity of total RNA were determined using the Nanodrop-2000 spectrophotometer, and 50 ng of total RNA was used as a template for each RT-PCR reaction. RT-PCR was performed using the SuperScript™ III One-Step RT-PCR System with Platinum™ Taq DNA Polymerase (Cat#12574018, ThermoFisher Scientific, Scoresby, VIC, Australia) according to the manufacturer’s instructions. A forward primer (anneals to exon 10), GTGGGCTACGAACTGAAGG, and a reverse primer (anneals to exon 13), GTCGCGGATGGTGGACTTC, were used for PCR reaction at an annealing temperature of 57 °C and amplification for 30 cycles. Following the RTPCR, the amplicons were fractionated on 2% agarose gels, and a 100 bp ladder was included for size comparison. The gel was then stained with 1X Redsafe (Cat#21141, iNtRON Biotechnology, Seongnam, Kyonggi-do, Republic of Korea) in TAE buffer for 10–15 min and subsequently washed for 20–30 min in ddH_2_O. The gel image was captured on a Vilber Lourmat Fusion FX system, using Fusion Capt software version 17.03, and ImageJ version 1.8.0_201 was used for image analysis and densitometry.

### 4.7. Quantitative Determination of PRPF31 Transcript

cDNA synthesis was performed using random primers with an SSIV cDNA reverse transcription kit (Sigma-Aldrich) from 100–200 ng of total RNA. Quantitative PCR was performed with specific primers for *PRPF31* (exon3F/Exon4R), a forward primer, GAGGAGACACAGCTGGATCTT, and a reverse primer, CCACGATGACGCGGTATTCAG, using Fast SYBR Green Master Mix (ThermoFisher Scientific, Scoresby, VIC, Australia). Reactions were set up in triplicate in 384-well plates with all amplified products ranging in size from 100 to 200 bp. The plates were run on a CFX384 Touch^TM^ Real-Time PCR detection system (Bio-Rad Laboratories Pty., Ltd., Gladesville, Australia). The PCR reaction was performed using the following conditions: activation of Taq polymerase and denaturation at 95 °C for 20 s followed by 40 cycles of 3 s at 95 °C and 30 s at 60 °C. The specificity of the amplified products was determined after analysis of the melting curve, carried out at the end of each amplification using one cycle at 95 °C for 15 s, then a graded thermal increase of 60 °C to 95 °C. *TBP* expression served as a housekeeping control using a forward primer, TCTTTGCAGTGACCCAGCATCAC, and a reverse primer, CCTAGAGCATCTCCAGCACACTCT. Normalized gene expression values against *TBP* were obtained using the 2^−ΔΔCT^ method.

### 4.8. Immunofluorescence Assay

Cells were seeded onto coverslips and, after AO transfection and incubation, were fixed with ice-cold acetone-methanol (1:1) and then blocked in 10% filtered goat serum (GS) in PBS containing 0.2% Triton-X (PBST). PRPF31 was detected with Rabbit anti-PRPF31 (Cat#HPA041939, Sigma-Aldrich) polyclonal antibody at a dilution of 1:100 in 1% GS/PBST for 1 h and the PRPF31 primary antibody was detected using Goat anti-Rabbit IgG (H+L) Secondary Antibody, Alexa Fluor™ 488 (1:400 dilution; Cat# A-11008, ThermoFisher Scientific, Scoresby, VIC, Australia). For nuclei detection, cells were stained with Hoechst 33342 (1 mg/mL diluted 1:125, Cat#B2261, Sigma-Aldrich) for 5 min at room temperature. Coverslips were then mounted onto glass slides using Prolong^TM^ Gold Antifade Mountant (Cat#P10144, ThermoFisher Scientific, Scoresby, VIC, Australia). Images of the cells were captured with a 40× objective on a Nikon TS100 microscope using NIS -Elements version AR 4.13.05 software.

## Figures and Tables

**Figure 1 ijms-25-03391-f001:**
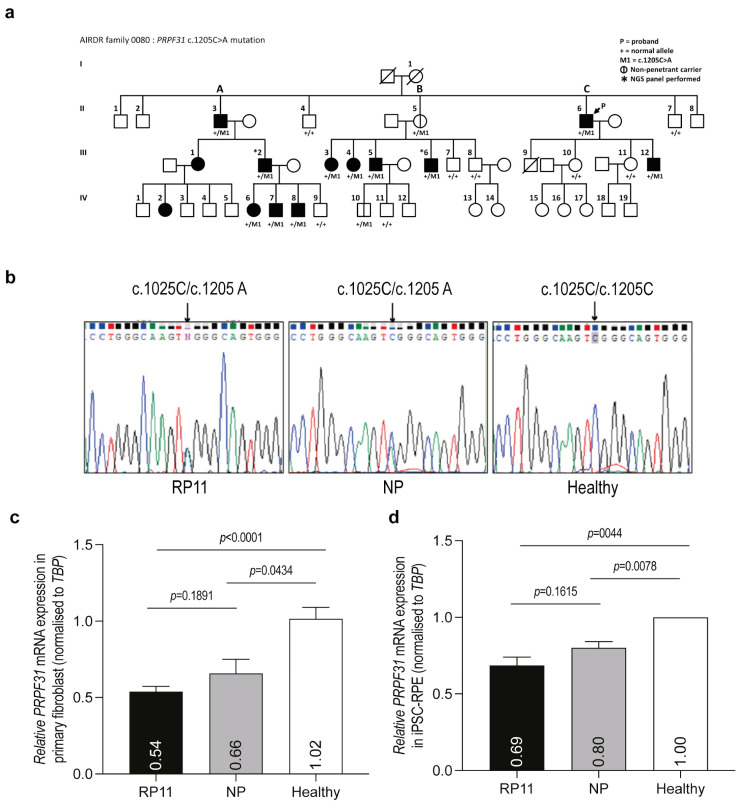
Characterization of *PRPF31* mutation and expression in RP11 patients and a non-penetrant family member within a large RP11 pedigree that shows incomplete penetrance. (**a**) RP11 pedigree with sample code used in this study (generations I–IV, families A, B, C). Black fill: affected RP11 patients with a confirmed diagnosis (*PRPF31* c.1205 C>A). Vertical line: non-penetrant subject. Arrow: proband. (**b**) Sanger sequencing chromatograms showing *PRPF31* sequence amplified from genomic DNA indicating the position of c.1205 (arrow) in an RP11 and a non-penetrant individual, compared to a healthy control. (**c**) RT-qPCR analysis of total *PRPF31* expression in fibroblasts from RP11 patients (*n* = 5 patients, c.1205 C>A), a non-penetrant family member (II5, c.1205 C>A), and an unaffected control. The expression of *PRPF31* was normalized to that of TATA-binding protein (*TBP*). *PRPF31* expression in the healthy control was set to 1. *n* = 3 independent experiments in triplicates were performed. Student’s *t*-test. (**d**) RT-qPCR analysis of total *PRPF31* expression in iPSC-derived RPE from an RP11 patient (III6, c.1205 C>A), a non-penetrant family member (II5, c.1205 C>A), and an unaffected, healthy subject. The expression of *PRPF31* was normalized to that of *TBP*. The *PRPF31* expression in the healthy subject (control) was set to 1. *n* = 3 independent experiments in triplicate were performed. Student’s *t*-test.

**Figure 2 ijms-25-03391-f002:**
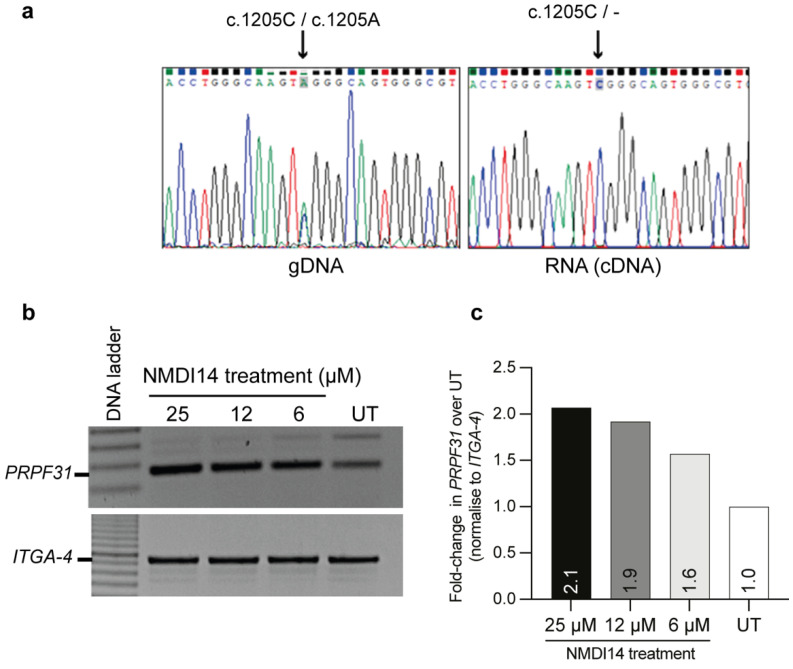
The mutation c.1205 C>A in *PRPF31* leads to mRNA degradation in the RP11-patient fibroblasts; the transcript was rescued by inhibiting nonsense-mediated mRNA decay. (**a**) PCR or RT-PCR products containing the variant were amplified from genomic DNA or cDNA derived from RP11 patient III6 in the pedigree and analyzed by Sanger sequencing. (**b**) Semi-quantitative RT-PCR analysis of *PRPF31* expression in the RP11 fibroblasts after treatment with NMDI14 for 16 h. Expression of integrin subunit alpha 4 (*ITGA-4*) served as a housekeeping control in the RP11 fibroblasts. (**c**) The intensity of RT-PCR amplicon bands was determined using ImageJ. The relative *PRPF31* expression was calculated from the intensity of the *PRPF31* product and normalized to that of *ITGA-4*. The *PRPF31* expression in untreated healthy control cells was set to 1 (*n* = 1).

**Figure 3 ijms-25-03391-f003:**
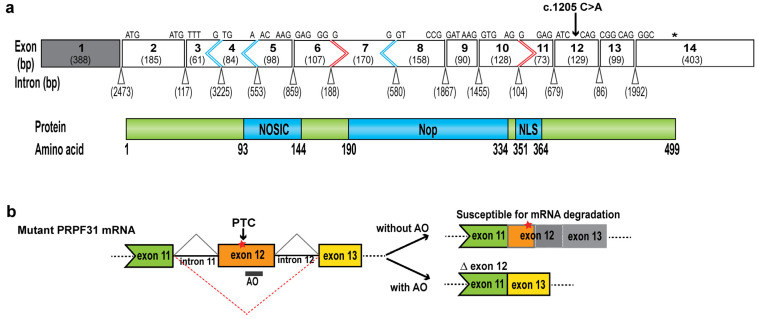
Schematic showing the splicing alteration strategy to bypass *PRPF31* exon 12 harboring a PTC (c.1205 C>A). (**a**) *PRPF31* includes 14 exons, with 13 exons encoding the protein of 499 amino acids. The shape of the boxes indicates the reading frame. In-frame exons are depicted by rectangles, whereas exons with junctions that interrupt codons are shown with chevron sides (blue, red). The number in the bracket in each exon indicates the exon length. Triangles between adjacent exons indicate the introns, intron length is shown in parentheses. The asterisk above exon 14 indicates the normal stop codon (TAA). The length of the PRPF31 protein is represented with functional domains at the stated amino acid positions, corresponding to the protein shown above the encoded exon. (**b**) AO design strategy to alter pre-mRNA splicing and induce exclusion of exon 12 harboring a PTC from the *PRPF31* transcript and generate a truncated *PRPF31* isoform, missing 129 nucleotides. The *PRPF31* c.1205 C>A allele transcript is shown to reflect the consequence the mutation on the transcript. The PTC is represented by a red asterisk.

**Figure 4 ijms-25-03391-f004:**
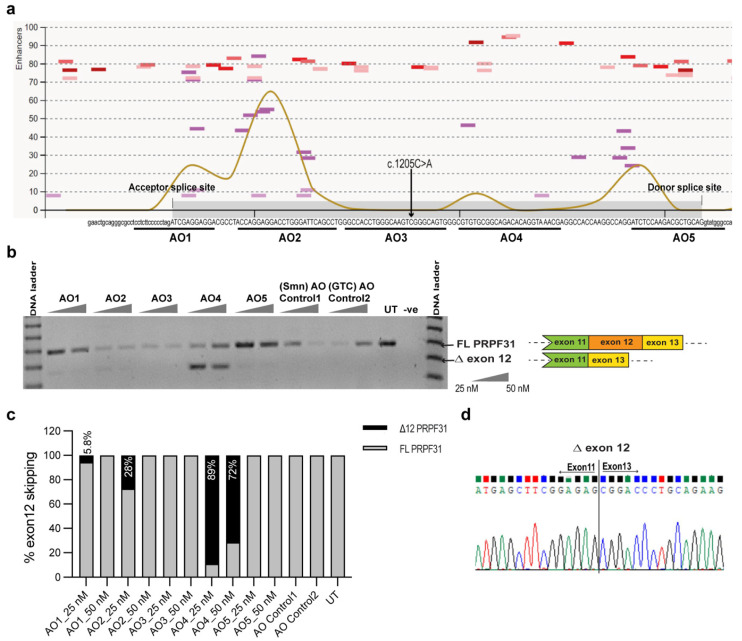
Design and screening of 2′*O*-methyl-PS AOs in patient fibroblasts. (**a**) Illustration of exon splice motifs predicted using the Human Splice Finder online tool and plotted along with the nucleic acid sequence of the PRPF31 exon 12 (grey bar) and corresponding flanking introns. Colored lines indicate motifs to which splicing enhancer proteins are predicted to bind. The brown line indicates the cumulative score of splice enhancers ranging from 0 to 100. Oligonucleotides, namely, AOs 1–5, and underlines are target binding sites of AOs. An arrow indicates the mutation site. (**b**) RP11 fibroblasts were transfected with 2′*O*-methyl PS AOs at indicated concentrations and incubated for 24 h; RT-PCR analysis showing exon 12 skipping (∆) and shorter amplicons due to the loss of 129 bp. FL: full-length transcript. (**c**) The percentage of exon 12 skipping and full-length PRPF31 transcript was analyzed using ImageJ software. The total PRPF31 transcript was set to 100%. (**d**) Sanger sequencing confirmed precise PRPF31 exon 12 skipping, and the new exon 11/exon 13 junction was induced by the AO4. GTC; GeneTools Control PMO.

**Figure 5 ijms-25-03391-f005:**
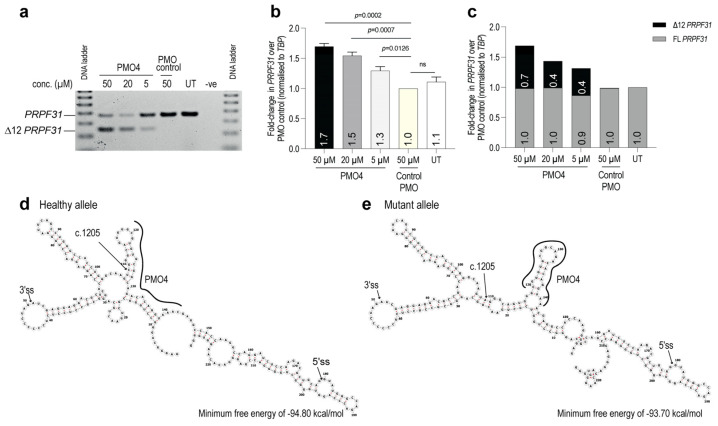
PMO-mediated alteration of *PRPF31* pre-mRNA splicing to bypass the c.1205 C>A nonsense mutation. (**a**) RT-PCR analysis of *PRPF31* showing exon 12 removal from the mRNA in PMO-treated fibroblasts (**b**) The relative expression of total *PRPF31* mRNA was determined using quantitative RT-PCR analysis. The expression of *PRPF31* mRNA was normalized to the *TBP* transcript. The level of *PRPF31* in GeneTools control PMO-treated cells was set to 1. FL: full-length transcript. *n* = 3 independent experiments were performed in triplicates, ns= not significant. Student’s *t*-test. (**c**) Relative expression of full-length and truncated *PRPF31* transcript using RT-qPCR analysis. The expression of *PRPF31* was normalized to that of *TBP*. *PRPF31* expression in untreated cells was set to 1. *n* = 1 biological replicate; *n* = 3 technical replicates. (**d**,**e**) The RNAfold online tool was used to predict the secondary structure of exon 12 plus 50 nucleotides of flanking introns for both the healthy and mutated transcripts. The arrow indicates the c.1205 position within exon 12 where the mutation occurred. The black line indicates the changes in the binding of PMO4 between the healthy and mutated transcripts.

**Figure 6 ijms-25-03391-f006:**
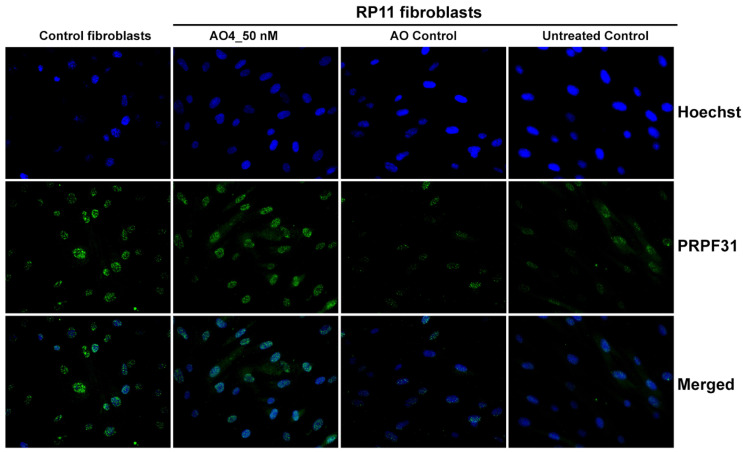
Intracellular localization of PRPF31 protein in healthy control and RP11 fibroblasts. Immunofluorescence staining of PRPF31 protein in control and patient fibroblasts, with or without splice-switching AO4 treatment. Cells were probed with rabbit anti-PRPF31 antibody against N-terminal PRPF31 (amino acid 1–100) and followed by Alexa488 anti-rabbit IgG (green) and counterstained with Hoechst (blue) to show nuclei. 40× magnification. Scale bar = 40 μm.

**Table 1 ijms-25-03391-t001:** List of splice-switching AO sequences designed to induce *PRPF31* exon 12 skipping and controls.

AO/PMO Name	AO Coordinate	AO Sequence (5′>3′)
AO1	PRPF31_H12A(−14+11)	UCCUCCUCGAUCUAGGGGGAAGAGG
AO2	PRPF31_H12A(+17+41)	AGGCUGAAUCCCAGGUCCUCCUGGU
AO3	PRPF31_H12A(+43+67)	CACUGCCCGACUUGCCCAGGUGGCC
AO4 ^1^	PRPF31_H12A(+70+94)	CGUUUACCUGUGUCUGCCGCACACG
AO5	PRPF31_H12D(+18−7)	CCCAUACCUGCAGCGUCUUGGAGAU
Control 1 AO	SMN H7A(+07+31)	ACCUUCCUUCUUUUUGAUUUUGUCU
PMO4 ^1^	PRPF31_H12A(+70+94)	CGTTTACCTGTGTCTGCCGCACACG
Control 1 PMO	SMN_H7A(+07+31)	ACCTTCCTTCTTTTTGATTTTGTCT
Control 2 PMO	GTC	CCTCTTACCTCAGTTACAATTTATA

^1^ The AO4 and PMO4 contain the same nucleotide sequence, with a U to T substitution.

## Data Availability

Data are available from the corresponding author and the first author upon request.

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
