# Peer review of "A Precision Therapy Approach for Retinitis Pigmentosa 11 Using Splice-Switching Antisense Oligonucleotides to Restore the Open Reading Frame of PRPF31"

_ijms, 2024, doi:10.3390/ijms25063391_

Round 1

Reviewer 1 Report

Comments and Suggestions for Authors

This is a very well carried research on a family carrying a incomplete penetrance mutation of PRPF31, together with a proposed mechanism for a future treatment.

The fact that the level of PRPF31 mRNA expression of 0.66 is sufficient for a normal phenotype (compared to 0.54 in RP11 patients) is encouraging. However, I do not think that performing Student's t test on a sample of 5 patients, one non-penetrant subject and one healthy control is enough in order to talk about statistical significance. I would skip the discission about statistical significance (p values presented in figures 1.c and 1.d and lines 239-240) and I think the scientific value of the paper will not decrease. 

Also, at line 221 it's missing the reference to figure 1.c.

The proposed use of antisense oligonucleotides in order to obtain a shorter but still effective protein is an elegant solution (and possibly useful in clinical practice, as shown in references 16-17.

The lack of correlation between the efficiency of AOs and the predicted ESE strength scores is notable (also reported by ref 28).

The authors have managed to obtain an AO (i.e. AO4) that efficiently induced exon 12 exclusion from the transcript, thereby increasing PRPF31 expression by a factor of 1.70

This might be clinically useful since the authors have proven that a small increase in expression may make the difference between normal and RP phenotype. 

This is a sound research with potential clinical relevance and I support its publication.

Author Response

Dear Reviewer

Thank you for taking the time to review our manuscript entitled " A precision therapy approach for retinitis pigmentosa 11 using splice-switching antisense oligonucleotides to restore the open reading frame of PRPF31" and for providing valuable feedback. We appreciate your thorough evaluation and insightful comments, which have undoubtedly helped strengthen the quality of our manuscript.

We have carefully considered each of your suggestions and criticisms, and we provide our responses below:

Comment:The fact that the level of PRPF31 mRNA expression of 0.66 is sufficient for a normal phenotype (compared to 0.54 in RP11 patients) is encouraging. However, I do not think that performing Student's t test on a sample of 5 patients, one non-penetrant subject and one healthy control is enough in order to talk about statistical significance. I would skip the discussion about statistical significance (p values presented in figures 1.c and 1.d and lines 239-240) and I think the scientific value of the paper will not decrease.

Response: We acknowledge the concern about our small sample size affecting the robustness of our statistical analysis. We agree with your suggestion to omit the discussion on statistical significance and have made the necessary revisions to ensure clarity and accuracy without compromising scientific value. Furthermore, we have revised this paragraph to improve the context, while ‘toning down’ the significance of differences in PRPF31 expression between RP11 and non-penetrance [See the revised paragraph,  lines 69-77].

Comment : At line 221 it's missing the reference to figure 1.c.

Response: Thank you for noting the missing reference to Figure 1.c;  line 221. We have addressed this oversight in the revised manuscript. 

Once again, we sincerely thank you for your constructive feedback and for guiding us in refining our manuscript. 

Reviewer 2 Report

Comments and Suggestions for Authors

The manuscript titled "A precision therapy approach for retinitis pigmentosa 11 using splice-switching antisense oligonucleotides to restore the open reading frame of PRPF31" presents a highly specialized and innovative research effort targeting the genetic basis of Retinitis Pigmentosa 11 (RP11) via splice-switching antisense oligonucleotides (AOs). This approach aims to correct the effects of truncating mutations in PRPF31, which are implicated in the disease's pathogenesis, by restoring the open reading frame of PRPF31 transcripts.

Strengths:

  • Innovative Approach: The therapeutic strategy utilizing splice-switching AOs represents a novel and targeted approach for treating RP11, a form of retinitis pigmentosa without available therapies.
  • Comprehensive Experimental Design: The study encompasses a thorough experimental workflow, from the characterization of PRPF31 expression in patient-derived cells to the design and efficacy testing of AOs.
  • Clear Demonstration of Concept: The results convincingly demonstrate that AO-induced exon skipping can effectively increase PRPF31 mRNA levels and potentially mitigate the haploinsufficiency driving RP11 pathology.

Areas for Improvement:

  1. Detailed Mechanistic Insights: While the study successfully demonstrates the feasibility of exon skipping to restore PRPF31 levels, further elucidation of how the truncated PRPF31 protein retains functionality would strengthen the manuscript. Specifically, exploring the functional implications of the missing amino acids on the protein's role in pre-mRNA splicing and retinal cell health could provide deeper insights into the therapeutic strategy's potential efficacy.
  2. Broader Applicability: The manuscript focuses on a specific PRPF31 mutation. Expanding the scope to include a broader range of PRPF31 mutations could enhance the study's relevance and applicability, given the genetic heterogeneity of RP11.
  3. Long-term Efficacy and Safety: The manuscript would benefit from a discussion on the anticipated long-term efficacy and safety of the proposed therapy, including potential off-target effects of AOs and strategies to monitor and mitigate these risks.
  4. Comparative Analysis with Other Therapies: A comparative discussion of this approach relative to other emerging therapies for retinitis pigmentosa, such as gene therapy or CRISPR-Cas9-based strategies, would provide valuable context for readers. Highlighting the advantages or complementary aspects of splice-switching AOs could underscore the significance of this research.
  5. Clinical Translation Pathway: Providing a clearer roadmap for clinical translation, including the steps required to move from in vitro studies to clinical trials, would be beneficial. This could include discussions on pharmacokinetics, delivery mechanisms, and regulatory considerations specific to splice-switching therapies.
  6. Data Presentation and Visualization: Enhancing the clarity and impact of figures and tables by simplifying complex data presentations and providing clearer, more detailed legends and annotations would improve the manuscript's readability and accessibility.

7.     References: Update the references to include the most recent studies in the field. In particular, consider including additional references to support the discussion and to provide context to the study’s findings. I suggest adding data related to recent bulk transcriptomics studies which could represent a strong substrate to enforce the role of described molecular mechanisms, such as the recent PMID: 36490268, PMID: 32560555 and PMID: 36593397.

Comments on the Quality of English Language

The English should be improved.

Author Response

Dear Reviewer

Thank you for taking the time to review our manuscript entitled " A precision therapy approach for retinitis pigmentosa 11 using splice-switching antisense oligonucleotides to restore the open reading frame of PRPF31" and for providing valuable feedback. We appreciate your thorough evaluation and insightful comments, which have undoubtedly helped strengthen the quality of our manuscript.

We have carefully considered each of your suggestions and criticisms, and we provide our responses below:

Comment : While the study successfully demonstrates the feasibility of exon skipping to restore PRPF31 levels, further elucidation of how the truncated PRPF31 protein retains functionality would strengthen the manuscript. Specifically, exploring the functional implications of the missing amino acids on the protein's role in pre-mRNA splicing and retinal cell health could provide deeper insights into the therapeutic strategy's potential efficacy.

Response: We acknowledge the importance of conducting functional studies in retinal cells, and we recognize that this aspect is a limitation of our current study. We appreciate your suggestion to explore this area further in future research. In the revised version of our manuscript, we emphasize this limitation and discuss the need for additional studies to elucidate the functional implications of the missing amino acids on the protein's role in pre-mRNA splicing and retinal cell health. By addressing this limitation, we aim to enhance the scientific robustness and relevance of our research.

 Comment: The manuscript focuses on a specific PRPF31 mutation. Expanding the scope to include a broader range of PRPF31 mutations could enhance the study's relevance and applicability, given the genetic heterogeneity of RP11.

Response: In response to your recommendation, we expand the discussion to other exons in PRPF31 that may be feasible for a similar intervention strategy. We focus on ‘in-frame’ exons that do not encode functional or binding domains, suggesting that their exclusion may not impact the localization or function of the protein. This analysis aims to provide a comprehensive understanding of potential therapeutic targets beyond the specific mutation studied in our manuscript. Please refer to lines 558-575 for the additional paragraph addressing this point.

 Comment: The manuscript would benefit from a discussion on the anticipated long-term efficacy and safety of the proposed therapy, including potential off-target effects of AOs and strategies to monitor and mitigate these risks.

Response: While we do not delve into the anticipated long-term efficacy and safety of the proposed therapy itself, as it is outside the scope of this study, we have expanded the discussion in the revised manuscript to address the promising long-term effectiveness of antisense oligonucleotides, a drug class that likely relevant to the readers. Additionally, we have included a reference to a detailed review of pre-clinical evaluation of the antisense oligonucleotide drug class in our discussion. See lines 585-590 that address this point.

Comment: A comparative discussion of this approach relative to other emerging therapies for retinitis pigmentosa, such as gene therapy or CRISPR-Cas9-based strategies, would provide valuable context for readers. Highlighting the advantages or complementary aspects of splice-switching AOs could underscore the significance of this research.

Response: We agree that providing context by highlighting the advantages or complementary aspects of splice-switching antisense oligonucleotides could underscore the significance of our research. In the revised manuscript, we include (i) an additional paragraph in the introduction to highlight the uniqueness of our approach compared to other recent studies, and (ii) a discussion of splice-switching AOs, relative to recent PRPF31 augmentation strategies including AAV gene transfer and PMO targeting CNOT3 to indirectly enhance PRPF31 expression. Please see details in the paragraph added at lines 544-557 in the manuscript.

Comment: Enhancing the clarity and impact of figures and tables by simplifying complex data presentations and providing clearer, more detailed legends and annotations would improve the manuscript's readability and accessibility

Response: The Figure legends have undergone minor adjustments to improve the interpretation of the respective figure. Furthermore, we have carefully reviewed the paper and implemented slight edits, with the aim to enhance its readability. We are confident that these changes contribute to the overall clarity of the manuscript.

Comment: Update the references to include the most recent studies in the field. In particular, consider including additional references to support the discussion and to provide context to the study’s findings. I suggest adding data related to recent bulk transcriptomics studies which could represent a strong substrate to enforce the role of described molecular mechanisms, such as the recent PMID: 36490268, PMID: 32560555 and PMID: 36593397.

 Response: While we appreciate your suggestion to include additional references, upon careful consideration, we have found that some of the references you provided may not be directly relevant to our study's focus. However, we incorporated recent, pertinent literature to support our discussion and provide context to our findings in the revised manuscript.

We believe that the revisions made in response to your comments have significantly strengthened the quality of our manuscript. We look forward to hearing your feedback on the revised manuscript.